# Feasibility of data linkage in the PARAMEDIC trial: a cluster randomised trial of mechanical chest compression in out-of-hospital cardiac arrest

Chen Ji,[1] Tom Quinn,[2] Lucia Gavalova,[2] Ranjit Lall,[1] Charlotte Scomparin,[1] Jessica Horton,[1] Charles D Deakin,[3,4] Helen Pocock,[4] Michael A Smyth,[1] Nigel Rees,[5] Samantha J Brace-McDonnell,[1,6] Simon Gates,[1,7] Gavin D Perkins[1,6]

For numbered affiliations see end of article.

**Correspondence to**
Dr Gavin D Perkins;
g.d.perkins@warwick.ac.uk

## ABSTRACT

**Objectives** There is considerable interest in reducing the cost of clinical trials. Linkage of trial data to administrative datasets and disease-specific registries may improve trial efficiency, but it has not been reported in resuscitation trials conducted in the UK. To assess the feasibility of using national administrative and clinical datasets to follow up patients transported to hospital following attempted resuscitation in a cluster randomised trial of a mechanical chest compression device in out-of-hospital cardiac arrest.

**Methods** Hospital data on trial participants were requested from Hospital Episode Statistics (HES), the Intensive Care National Audit and Research Centre, and Myocardial Ischaemia National Audit Project and National Audit of Percutaneous Coronary Interventions, using unique patient identifiers. Linked data were received between June 2014 and June 2015.

**Results** Of 4471 patients randomised in the pre-hospital randomised assessment of a mechanical compression device in cardiac arrest (PARAMEDIC) trial, 2398 (53.6%) were not known to be deceased at emergency department arrival and were eligible for linkage. We achieved an overall match rate of 86.7% in the combined HES accident and emergency, inpatient and critical care dataset, with variable match rates (4.2%–80.4%) in individual datasets. Patient demographics, cardiac arrest-related characteristics and major outcomes were predominantly similar between HES matched and unmatched groups, in the linkage apart from location, response time and return of spontaneous circulation (ROSC) at handover.

**Conclusions** This study shows that it is feasible to track patients from the prehospital setting through to hospital admission using routinely available administrative datasets with a moderate to high degree of success. This approach has the potential to complement the trial data with the demographic and clinical management information about the studied cohort, as well as to improve the efficiency and reduce the costs of follow-up in cardiac arrest trials.

**Clinical trial registration** ISRCTN08233942; Post-results.

## BACKGROUND

Well conducted and reported randomised controlled trials (RCTs) are considered the gold

---

**Strengths and limitations of this study**

► First study evaluating the supplement of routinely collected administrative data in a cardiac arrest trial in the UK.
► Data linkage was made to different UK national registries.
► The matching reliability was suboptimal due to relaxed matching criteria, matching method and possible data quality issues.
► Routine data were not fully available for all trial patients transported to hospital.
► The findings of our study are not generalisable to facilitate trial recruitment since it was considered unrealistic in the clinical context of cardiac arrest.

---

standard in evaluation of new or established clinical interventions. In cardiac arrest resuscitation science, only a small minority (1%) of contemporary international guideline recommendations are based on the highest level of evidence from more than one RCT, meta-analysis of high quality RCTs or RCTs corroborated by high-quality registry studies.[1] High-quality trials to address outcomes of interest to patients following cardiac arrest (eg, long-term survival, neurocognitive status and disability)[2] are complex, labour intensive and expensive to perform. Many studies in cardiac arrest are therefore too small or inadequately conducted (with a predominance of observational studies which are prone to bias) to provide reliable estimates of treatment effect or harm to patients. Consequently, for the majority of resuscitation interventions, there is a paucity of high-quality evidence. Funders (typically government agencies) have called for proposals for low-cost, more efficient trials.[3]

Traditional trial methods of patient tracking and data access in individual hospitals is challenging with limited resources.

Cardiovascular medicine has attempted to improve the efficiency of the trial design by pioneering the concept of registry-based randomised trials, using clinical quality registries and administrative datasets. In the Thrombus Aspiration during ST-Segment Elevation Myocardial Infarction (TASTE) trial, undertaken in Sweden, both patient enrolment and follow-up were conducted using the Swedish Web System for Enhancement and Development of Evidence-based Care in Heart Disease Evaluated According to Recommended Therapies registry.[4] On publication, this registry-based trial was hailed as the '*next disruptive technology'* in clinical research and as a new clinical trial paradigm.[5 6] Subsequent registry-based trials have been reported in a comparison of radial versus femoral access in women undergoing percutaneous coronary intervention in the USA[7] and of supplemental oxygen versus ambient air in patients with suspected acute myocardial infarction in Sweden.[8]

To our knowledge, however, there are no reports of registry-based randomised trials in resuscitation science. However, should accessing registry data to ascertain outcomes in a prehospital cardiac arrest trial (eg, length of stay/patient pathways/survival status) to be feasible, this could be one way of significantly improving efficiency and reducing costs of conducting high-quality randomised trials in resuscitation.

In the PARAMEDIC trial, the in-hospital data collection process was complex, expensive and labour intensive, with research paramedics visiting multiple hospitals across large geographical areas to extract data from hospital records. Patients transported to hospital following resuscitation from cardiac arrest follow multiple clinical pathways depending on their clinical status and treatments. As hospital data are routinely collected and managed by national registries, using these registries could save resources and time in the in-hospital data collection and potentially reduce the burden on patients and relatives in the sensitive period following cardiac arrest.

This paper reports our assessment of the feasibility of linking data collected for the purposes of patient follow-up in a pragmatic, cluster RCT of a mechanical chest compression device undertaken in the UK prehospital setting, with large national administrative and specialist registries.

## METHODS

The PARAMEDIC trial examined the effectiveness of LUCAS-2, a mechanical chest compression device, in 4471 patients with out-of-hospital cardiac arrest (OHCA). The study was a cluster randomised trial whereby emergency medical service (EMS) vehicles were randomised to carry the LUCAS-2 device (intervention) or not (control). Full details of the trial protocol have been published previously.[9] In summary, adults with OHCA where resuscitation was attempted by EMS personnel and attended by a trial vehicle were included. Patients with traumatic cardiac arrest or suspected to be pregnant were excluded. Trial recruitment ran from 15 April 2010 to 10 June 2013.

We have previously reported primary outcome (30-day survival),[10] secondary outcomes,[11] an economic analysis[12] and characteristics of patients who were not resuscitated.[13]

### Data sources

The PARAMEDIC trial used four sources of data that were linked to the trial dataset: UK National Health Service (NHS) Hospital Episodes Statistics (HES), Myocardial Ischaemia National Audit Project (MINAP),[14] National Audit of Percutaneous Coronary Interventions (NAPCI)[15] and Case Mix Programme (CMP)[16] to obtain data on hospital stay and treatment or procedures that trial patients received in hospital.

We used the MINAP, NAPCI and CMP data for the health economic analysis[12] and long-term postadmission outcomes[11] and to validate the hospital length of stay or stay in the intensive care (secondary outcomes for the efficacy part of the trial) and also to gain insight into the specifics of the treatment or procedures that trial patients received during their hospital stay. Characteristics of the registries are summarised in table 1.

### Patient population

Patients (denominator) for this linkage study were patients from the PARAMEDIC trial who were transported to hospital by EMS and not known to be deceased (ie, documented as alive or unknown status) on arrival at the emergency department (ED).

Since NHS Digital (NHSD), responsible for HES, only provides annual data up to 1st April each year, no data on trial patients recruited on or after 1 April 2013 had any HES data returned for this data request. We therefore limited our analysis of the linked registry data to patients recruited to the PARAMEDIC trial between April 2010 and March 2013.

### Study approvals

The PARAMEDIC trial was sponsored by the University of Warwick, UK. The study was conducted in accordance with the principles of Good Clinical Practice and the Mental Capacity Act (2005). Specific approval for access to personal data without consent and the data linkage reported in this paper was obtained from the Confidentiality Advisory Group, part of the Health Research Authority (reference: ECC 2–02 (c)/2011). At the time of the study, this activity was undertaken by the National Information Governance Board for Health and Social Care Ethics and Confidentiality Committee.

### Patient and public involvement

Patient and public representatives were invited to the Trial Steering Committee meetings during the development and conduct of the main trial. They agreed with the data collection via linkage to reduce the burden on patients and relatives. They were regularly informed of this study and other trial outputs. The results of this study will be disseminated in different ways, including presentation on the publicly accessible trial webpage.

**Table 1** Characteristics of registries, participation and case ascertainment

| Registry/dataset | Source | Description | Participation and case ascertainment* during the trial period |
|---|---|---|---|
| Paramedic trial | Warwick Clinical Trials Unit | Trial patient cohort that survived admission to a hospital. | N/A |
| Hospital Episode Statistics | NHS Digital | Collection of information on all NHS hospital inpatients, accident and emergency, critical care and outpatients that enables healthcare providers to be paid according to their levels of activity. | All hospitals. Case ascertainment 100%. |
| Case Mix Programme | Intensive Care National Audit and Research Centre | Audit of patient outcomes from all adult, general critical care units in England, Wales and Northern Ireland. Other specialist units, including neurosciences, cardiac and high dependency units, also participate. | Over 90% of critical care units. Case ascertainment not reported. |
| Myocardial Ischaemia National Audit Project | National Institute for Cardiovascular Outcomes Research (NICOR) | National audit of patients with acute coronary syndrome admitted to all hospitals in England, Wales and Northern Ireland. Data are collected prospectively at each hospital by secure electronic system, electronically encrypted and transferred online to a central database. | All hospitals. Case ascertainment not reported. |
| National Audit of Percutaneous Coronary Interventions | NICOR | National audit of all percutaneous coronary intervention (PCI) procedures from NHS and non-NHS hospitals in the UK. | All hospitals. Case ascertainment 97%. |

*Case ascertainment – rate (eg, %) of eligible cases included in a registry/database.
NHS, National Health Service.

## Data linkage procedure

Data access applications were submitted to national administrative and disease registries between 2012 and 2014 to request patient case mix and clinical variables (online supplementary table 1). The following patient identifiers were sent to the NHSD, Intensive Care National Audit and Research Centre (ICNARC) and National Institute for Cardiovascular Outcomes Research (NICOR) to identify their clinical records: trial number, cardiac arrest date, ambulance service case number, 999 call time, hospital name, hospital arrival time, hospital handover time, patient name, NHS number, home address and postcode. The trial data were linked to the two NICOR datasets (MINAP and NAPCI) on two separate occasions by a different member of NICOR staff, which reassuringly generated the same results. Extracted anonymous data were encrypted and sent back to the trial team between June 2014 and June 2015.

Linked data may contain multiple, non-event-related hospital records within the requested linkage period. We first used patient cardiac arrest (trial event) date to identify the records with exactly matched admission/visit date in the respective data sources. However, event and admission dates could be different due to potential data definition discrepancies. For instance, a trial event could occur before midnight, and the patient was admitted to hospital after midnight. Therefore, we relaxed the date match criterion to a 5-day range (date of cardiac arrest with ±2 days). A matched record was redefined as if the admission/visit date falls in the range. We considered the range would be sufficiently large to mitigate against

any date discrepancies in different sources and also be reasonably small to reduce the chance of mismatch in the case of early readmission. Where multiple records could be matched to a single trial event in the same routine dataset, separate rules were used to extract the retrieved information: (1) where a patient had multiple episodes in HES, only the one with recorded death or discharge date was retained. If a patient had not been discharged from hospital, the episode with latest ward admission date was used. (2) Where multiple admissions to intensive care unit (ICU) were recorded in CMP, only the first ICU admission was linked to a trial event. (3) Since the MINAP dataset provided to us by NICOR only contained year and month of admission, only the earliest admission was used. (4) Only the first procedure was included for the linkage to the NAPCI registry data, since patients can have more than one interventional procedure (and thus another record) during the index admission.

## Data linkage rate

For HES data, we developed the linkage and match rate for linked and matched (or correctly linked) cases as follows:

$$\text{HES linkage rate} = \frac{\text{N of patients with linked HES inpatient, Critical care or A\&E data}}{\text{N of patients not known to be deceased at ED}}$$

$$\text{HES match rate} = \frac{\text{N of patients with matched (correctly linked) HES inpatient, Critical care or A\&E data}}{\text{N of patients not known to be deceased at ED}}$$

### N of patients not known to be deceased at ED

Similar equations were used to determine the rates for each of the datasets, that is, MINAP, NAPCI and CMP. As we were not able to confirm which patient should actually be collected in these datasets, we employed same denominator used in the above equations.

### Data linkage quality

Match rank is an indicator used in HES to show the confidence of match: 1 suggests the best match and 8 suggests the worst. Levels 1–3 appear to be of high quality as cases are matched based on a combination of unique NHS number and data of date of birth, sex and home postcode. The quality of linkage in matched HES was therefore summarised on the basis of percentage of levels 1–3.

### Data representativeness

Data representativeness was assessed in two comparisons. The first comparison intended to assess whether the patients with correctly linked (ie, matched) HES data could be representative of the trial population. It was carried out in patients with and without matched HES Inpatient, Critical Care or A&E data (comparison 1). The second comparison intended to assess the difference between two critical care data sources. We were not able to compare data from these two sources directly as some patient care data were collected in both databases. Hence, we split the patients by their linked data sources and made the comparison between patients with HES Critical Care only, with CMP data only and with both HES Critical Care and CMP data (comparison 2).

For both comparisons, we compared patient and event characteristics between the datasets. Continuous variables were compared using Mann-Whitney U test in comparison 1 and Kruskal-Wallis test in comparison 2. Categorical variables were compared using the $\chi^2$ test. A two-sided p value <0.05 was considered statistically significant. All analyses were conducted in SAS V.9.3 (SAS Institute, Cary, North Carolina, USA).

### Data security and destruction

We followed the Warwick Clinical Trials Unit Standard Operating Procedures for data storage, transfer and data sharing. The data were retained and destroyed in accordance with relevant regulations and the University of Warwick's Data Sharing Agreements.

### RESULTS

In the PARAMEDIC trial, 2695 patients were transported to hospital and not known to be deceased at ED. Of these, 2398 (89.0%) were recruited between April 2010 and March 2013 and were therefore included in this study (referred to as 'linkage patients'). The data requests to NHSD, ICNARC and NICOR retrieved different numbers of patient clinical records.

### Summary of the linkage

The flow chart of the linkage to HES is shown in figure 1. The linkage patients were grouped into ICU admitted (patients with matched HES Critical Care data) and not admitted (patients with other matched HES data). Meanwhile, patients with matched CMP data were also summarised in the flow chart. This presented a comparison between CMP and HES Critical Care. Three hundred and three patients were matched in both CMP and HES Critical Care. Overall, the linkage to HES data achieved a match rate of 86.7% (2079 of 2398) with allowed variation in dates (date of cardiac arrest with ±2 days), slightly improved from the use of exact date match approach (84.1%).

Linkage quality was high in matched cases: levels 1–3 accounted for 97.9%. In unmatched cases, 91.5% (292 of 319) had no linked HES data, and the rest, while linked with non-trial even related data, had a good match rank (≤3).

The summary of linkage and match rate in each dataset are shown in table 2. All datasets contained multiple linked records, indicating some patients had been linked to multiple admissions with possible multiple episodes. Among the 2398 linkage patients, individual match rate varied depending on the hospitalisation stage and received treatments. HES A&E had the highest individual match rate (80.4%). In the patients admitted to ICU, CMP provided 53 more matched patients with a lower proportion of unmatched data in linked patients compared with HES Critical Care.

A summary of retrieved information for each linked dataset as well as the degree of data missingness for each field is available in the online supplementary materials. In online supplementary table 2, the trial patients that had not been matched to the HES records were similar to those that with matched records in age (mean age 71.8 and 73.6, respectively), male (67.4% and 63.3%, respectively, were male). They were also similar between groups in initial cardiac arrest aetiology where most were of cardiac origin (85.3% and 85.9%, respectively) and in initial rhythm (shockable rhythm; 31.0% and 31.3%). Patients with unmatched data were more likely to have had a cardiac arrest in a public place (27.9%) compared with of those with matched records (16%), witnessed by bystander (53.3% vs 46.3%) and had longer EMS response time (7.2 min vs 6.1 min). Online supplementary table 3 illustrates the comparison of demographic and event characteristics of patients with matched HES Critical Care and CMP data. Characteristics were similar in all three groups, except for a significant difference in the EMS response time.

### DISCUSSION

This study aimed to demonstrate the feasibility of collecting trial outcome data during patient follow-up in a prehospital cardiac arrest trial via linkage to national registries. We achieved an overall match rate of 86.7% in

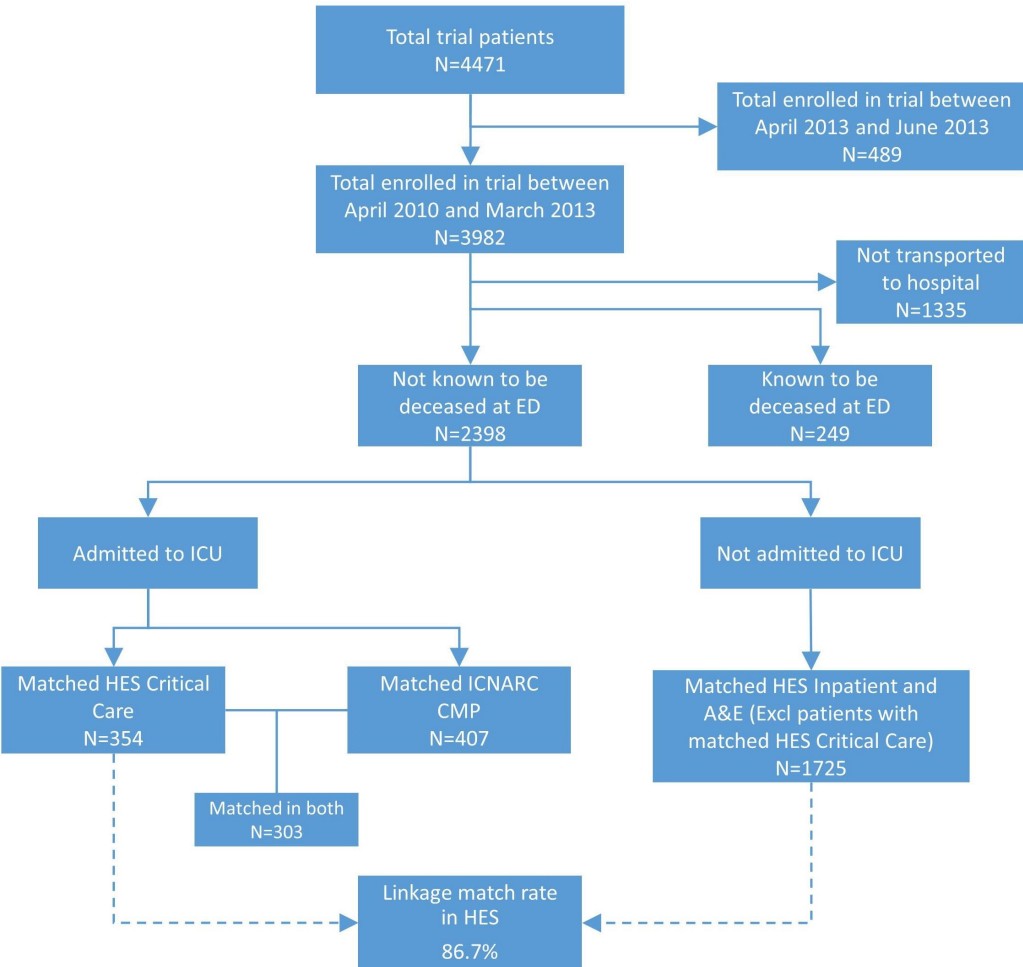

**Figure 1** Linkage match rate and flowchart of patients retrieving HES or CMP data. A&E, accident and emergency; CMP, Case Mix Programme; ED, emergency department; HES, Hospital Episodes Statistics; ICNARC, Intensive Care National Audit and Research Centre; ICU, intensive care unit.

2398 patients using HES data. The data linkage provided important administrative and additional clinical data that allowed extended analyses of the intervention effect and provided more details of patient journey in the trial. We also evaluated the representativeness of retrieved HES and CMP data by comparing patient and trial event characteristics. No substantial difference was found in patients with and without matched HES Inpatient, Critical Care or A&E data, as well as in patients with matched HES Critical Care only, CMP only and both datasets.

This was the first study evaluating the supplement of routinely collected administrative data in a cardiac arrest

**Table 2** Summary of linked PARAMEDIC trial patients to the respective registry databases

| Data source | Dataset | Number of linked records | Number of linked patients (linkage rate)* | Number of matched patients (match rate)* |
|---|---|---|---|---|
| NHSD | HES Inpatient | 12 875 | 1617 (67.4) | 771 (32.2) |
| | HES Critical Care | 545 | 433 (18.1) | 354 (14.8) |
| | HES A&E | 6434 | 2186 (91.2) | 1927 (80.4) |
| | Overall | 19 854 | 2277 (95.0) | 2079 (86.7) |
| ICNARC | CMP | 435 | 410 (17.1) | 407 (17.0) |
| NCAP | MINAP | 244 | 218 (9.1) | 182 (7.6) |
| | PCI | 153 | 128 (5.3) | 101 (4.2) |

*Percentage is calculated using the number in the column divided by 2398 linkage patients.
A&E, accident and emergency; CMP, Case Mix Programme; HES, Hospital Episodes Statistics; ICNARC, Intensive Care National Audit and Research Centre; NCAP, National Cardiac Audit Programme; MINAP, Myocardial Ischaemia National Audit Project; NHS, National Health Service; PCI, Percutaneous Coronary Intervention; NHSD, NHS Digital.

trial in the UK. Our match rate was in line with observational studies linking EMS data to hospital records[17–20] and data validation studies.[21] Our experience suggests it is feasible to obtain relevant data from administrative databases in a cardiac arrest trial. In addition to the high match rate reported in this paper, the matched data are deemed to be sufficiently representative of the trial population. The comparison between patients with and without matched HES showed low level of imbalance of event characteristics. We have found similar results in the matched ICU data.

The unmatched cases were likely to be associated with missing or inaccurate data. Data quality could be at increased risk due to the challenging circumstances of cardiac arrest and complexity of patient handling following hospital arrival. In addition, routine data in the chosen registries are not systematically adjudicated. Lack of clinical engagement may compromise the case ascertainment and data quality,[22] leading to suboptimal linkage. NHSD employs deterministic and probabilistic methods in the data linkage. The latter calculates probability weight based on combinations of linkage variables and determines linkage based on a cut-off threshold. Although this method largely improves the linkage, it could incorrectly link record pairs and miss valid ones, undermining the reliability of linkage.

Linkage to individual routine datasets resulted in variable match rates. HES A&E generated the highest rate of 80.4% as most patients were taken by EMS to ED for assessment before being admitted to specialist hospital units. Other rates reflected the proportion of specific groups of patients in the linkage. The CMP, MINAP and NAPCI registries are focused on selected patients with a specific diagnosis and/or requiring specialist care, reflected in strict inclusion and exclusion criteria; for example, MINAP comprises data on patients with suspected and/or confirmed acute coronary syndrome, NAPCI on interventional cardiology, while CMP registry collects data on patients admitted to critical care/ICUs within any given hospital. In this study, MINAP and NAPCI generated 9.1% and 5.3%, respectively. Patients who die in the ED are less likely to be recorded on MINAP, and only those patients receiving interventional cardiology are recorded in NAPCI.

Use of routine data has the potential to reduce the costs of conducting trials. The cost of the TASTE trial was reported as US$300 000, or approximately $50 per patient,[4] 2% of the cost of a traditional randomised trial, but differs from the PARAMEDIC trial in that we did not use registry data to identify and recruit patients in the challenging and time-pressured setting of OHCA. In the West of Scotland Coronary Prevention trial, data linkage reduced costs of long-term follow-up to less than 1% of trial budget.[23] However, the time cost of linkage could be unrealistic for some trials. Linkage for the PARAMEDIC trial took up to 3 years from application to the trial team obtaining the data. It has been suggested that NHSD, which performed the linkage to HES for our study, was overwhelmed with data linkage applications.[24] This may limit the usefulness of administrative data in trials with funder-imposed deadlines for completion.

## Limitations
Our study had several limitations. The matching reliability was suboptimal due to relaxed matching criteria (using range of event date), matching methods and potential issues of data quality and completeness, common to administrative data. Bohensky et al[25] conducted an evidence synthesis of data linkage studies and identified factors such as suboptimal or incomplete linkage leading to systematic bias. They considered the participant or population characteristics that can influence the validity and completeness of data linkage and may in turn lead to systematic bias in reporting. They reported variation in quality of data linkage across geographical/hospital sites, which could be due to high staff turnover or not sufficient resources allocated to the data collection and/or coding. We have not considered such variations in this study, but overall match quality was high in the matched cases.

Second, routine data were not fully available for all patients transported to hospital. Some patients were not included in the linkage as their data were not available in HES at the time of our data application. Although no substantial bias was shown, the generalisability of results could be limited. Several data fields were incomplete; for example, MINAP captures most ST elevation myocardial infarction (STEMI) cases, but data for non-STEMI are less complete. We also cannot confirm how many patients required specialist care and should be included in non-HES datasets. Therefore, we were unable to assess and report the impact of unmatched cases in in the linkage to these registries.

Third, we used the first matched admission without considering repeated or later admissions. We were therefore unlikely to fully describe patients' hospital pathway based on matched information.

Fourth, our focus for the present study was on assessing the feasibility of using administrative data for purposes of follow-up. We did not assess the utility of administrative data to facilitate recruitment of trial patients since this was considered unrealistic in the clinical context of cardiac arrest.

Fifth, we did not assess the financial cost of manual data collection at hospitals to compare with the cost of the use of registries in the trial linkage.

## Recommendations
Based on our experience, we made the following recommendations to improve the use of data linkage in trials:
1. When planning a trial using linkage to administrative registries, careful planning is required to assess availability of the required data. Linkage to routine data in different jurisdictions or multiple registries requires separate applications for data release and may be subject to data availability.

2. Trialists need to be mindful of prolonged processes for regulatory approvals, data release and validation. These processes may extend beyond trial funding.

3. Data linkage is a lengthy often unpredictable process in the application stage, possibly due to the restricted capacity of registries funded primarily to assess quality of care. Most registries in the NHS are funded as national audits and do not have sufficient resources for the timely processing of data sharing requests.

4. The quality of routinely collected data in the national registries may be inferior to that collected using traditional trial processes. Registry data are collected in high volume with limited resources, and the validation process is unlikely to be as robust as in trials that are better resourced. Moreover, collected variables in registries are reviewed periodically and may change to reflect advances in clinical practice, which can impact on data completeness. Therefore, we suggest that trialists use registry data as the main source of all in-hospital data points and active data collection by a study team as an auxiliary approach to collect data for the unmatched patients.

5. It is common for registries to charge a fee for data release, which should be costed in to trial budgets.

## CONCLUSIONS

This study shows that it is feasible to track patients from the prehospital setting through to hospital admission using routinely available administrative datasets with a moderate to high degree of success. This may improve the efficiency and reduce the costs for longer term follow-up in cardiac arrest trials.

**Author affiliations**
[1]Warwick Clinical Trials Unit, University of Warwick, Coventry, West Midlands, UK
[2]Faculty of Health, Social Care & Education, Kingston University and St George's, University of London, London, UK
[3]NIHR Southampton Respiratory Biomedical Research Unit, University Hospital Southampton NHS Foundation Trust, Southampton, UK
[4]South Central Ambulance Service NHS Foundation Trust, Otterbourne, UK
[5]Welsh Ambulance Service NHS Trust, Cardiff, UK
[6]Heart of England NHS Foundation Trust, Birmingham, UK
[7]Cancer Research UK Clinical Trials Unit, University of Birmingham, Birmingham, UK

**Acknowledgements** This is a summary of independent research partly funded by the National Institute for Health Research's (NIHR) Health Technology Assessment Programme (Grant Reference Number HTA-07/37/69). GDP received support as an NIHR Senior Investigator. MS is supported as an NIHR Doctoral Research Fellow. We would like to thank the independent members of the Trial Steering Committee (Jon Nicholl, Helen Snooks, Fionna Moore, Alasdair Gray, Martyn Box, Father Neil Bayliss (PPR), and John Long (PPR)) and the Data Monitoring Committee (Marion Campbell, Jerry Nolan and Kathy Rowan). We acknowledge support from the OHCAO registry project that is funded by the British Heart Foundation and Resuscitation Council (UK).

**Contributors** CS, JH, SG and GDP led on the data linkage. CJ analysed data. CJ, TQ and LG drafted the manuscript with input from the coauthors; RL, CS, JH, CDD, HP, MAS, NR, SJB-M, SG and GDP reviewed the manuscript. All authors approved the final version.

**Funding** The PARAMEDIC project was funded by the National Institute for HealthResearch's Health Technology Assessment programme (Grant Reference Number HTA-07/37/69).

**Disclaimer** The views expressed are those of the author(s) and not necessarily those of the NHS, the NIHR and the Department of Health.

**Competing interests** GDP, RL, TQ, CDD and SG report grants from NIHR HTA Programme during the conduct of the study. The other authors declare no competing interests.

**Patient consent** Obtained.

**Ethics approval** The PARAMEDIC trial was approved by the Coventry Research Ethics Committee (reference 09/H1210/69).

**Provenance and peer review** Not commissioned; externally peer reviewed.

**Data sharing statement** No additional data are available.

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
