## [Reviewer comments · BMJ Open]

ARTICLE DETAILS

TITLE (PROVISIONAL)	Feasibility of data linkage in the PARAMEDIC Trial: a cluster randomised trial of mechanical chest compression in out of hospital cardiac arrest
AUTHORS	Ji, Chen; Quinn, Tom; Gavalova, Lucia; Lall, Ranjit; Scomparin, Charlotte; Horton, Jessica; Deakin, Charles; Pocock, Helen; Smyth, Michael; Rees, Nigel; Brace-McDonnell, Samantha; Gates, Simon; Perkins, Gavin

VERSION 1 – REVIEW

REVIEWER	Stefan James Uppsala University Uppsala Sweden
REVIEW RETURNED	26-Jan-2018

GENERAL COMMENTS	I applaud the authors for having performed this study to investigate the use and validity of collecting outcome variables through alternative sources. I found the paper well written, informative and balanced. I have no concerns and recommend acceptance as is
--

REVIEWER	Sten Rubertsson Dept. of Surgical Sciences/Anaesthesiology & Intensive Care Medicine, Uppsala University, Sweden
REVIEW RETURNED	14-Feb-2018

GENERAL COMMENTS	This is an important paper describing the feasibility of data linkage between trial data and national registries of a resuscitation trial in the UK. This was done to explore the potential of future usage in the research field since large randomized trials are labor intense, burden with high costs but also to some extent with missing data. The manuscript is well written and clearly describes the potential but also the limitations. It identifies several issues that need to be considered for the researchers when planning for similar new studies. What would have been interesting for the reader is if any discussions have been performed with the authorities on how to use this experience in the future. Even if an overall match of 86.7 % is good, the precision is questionable and could bias the results. However, this was just a first effort in doing this linkage. Even if this is a national-UK analyze of the potential of linkage, I do think it gives the reader within the research field but also outside the field some important information that could be generalized.
--

REVIEWER	Robert W. Neumar University of Michigan, United States
REVIEW RETURNED	13-Apr-2018

GENERAL COMMENTS	This study evaluates the feasibility of using hospital-based registries
---

	to supplement treatment and outcome data collection for an out-of-hospital cardiac arrest (OHCA) clinical trial. This study is important in that it examines a mechanism to increase feasibility and reduce costs of OHCA clinical trials. Overall the approach accurately linked 86.7% of study patients not known to be deceased at time of hospital arrival. The methods are adequately described and important limitations are addressed. The conclusions are supported by the results. The manuscript could be strengthened by addressing several questions/comments below. Patients with unmatched data were more likely to have had a cardiac arrest in a public place, witnessed by a bystander, and longer EMS response time. Since these are all associated with outcome, how did the authors arrive at the conclusion that there was not significant selection for bias among the unmatched cohort? Although the focus of the study is feasibility, it would be helpful for the authors to discuss what specific OHCA data points can reliably be obtained from registry data and what data points are likely to require active data collection by a study team.
--	--

VERSION 1 – AUTHOR RESPONSE

Reviewer(s)' Comments to Author:

Reviewer: 1

Reviewer Name: Stefan James

Institution and Country: Uppsala University Uppsala Sweden Please state any competing interests: None

Please leave your comments for the authors below

I applaud the authors for having performed this study to investigate the use and validity of collecting outcome variables through alternative sources. I found the paper well written, informative and balanced. I have no concerns and recommend acceptance as is

RESPONSE: We appreciate very much the reviewer for the insightful comments.

Reviewer: 2

Reviewer Name: Sten Rubertsson

Institution and Country: Dept. of Surgical Sciences/Anaesthesiology & Intensive Care Medicine, Uppsala University, Sweden Please state any competing interests: None declared

Please leave your comments for the authors below

This is an important paper describing the feasibility of data linkage between trial data and national registries of a resuscitation trial in the UK. This was done to explore the potential of future usage in the research field since large randomized trials are labor intense, burden with high costs but also to some extent with missing data. The manuscript is well written and clearly describes the potential but also the limitations. It identifies several issues that need to be considered for the researchers when planning for similar new studies. What would have been interesting for the reader is if any discussions have been performed with the authorities on how to use this experience in the future. Even if an overall match of 86.7 % is good, the precision is questionable and could bias the results. However,

this was just a first effort in doing this linkage. Even if this is a national-UK analyze of the potential of linkage, I do think it gives the reader within the research field but also outside the field some important information that could be generalized.

RESPONSE: We appreciate very much the reviewer for the insightful comments.

Reviewer: 3

Reviewer Name: Robert W. Neumar

Institution and Country: University of Michigan, United States Please state any competing interests:

None declared

Please leave your comments for the authors below

This study evaluates the feasibility of using hospital-based registries to supplement treatment and outcome data collection for an out-of-hospital cardiac arrest (OHCA) clinical trial. This study is important in that it examines a mechanism to increase feasibility and reduce costs of OHCA clinical trials. Overall the approach accurately linked 86.7% of study patients not known to be deceased at time of hospital arrival. The methods are adequately described and important limitations are addressed. The conclusions are supported by the results. The manuscript could be strengthened by addressing several questions/comments below.

Patients with unmatched data were more likely to have had a cardiac arrest in a public place, witnessed by a bystander, and longer EMS response time. Since these are all associated with outcome, how did the authors arrive at the conclusion that there was not significant selection for bias among the unmatched cohort?

RESPONSE: We thank the reviewer for raising this question. We have removed this statement in the main paper: "Overall the results suggested no substantial bias." and modified the abstract as follows: "No strong evidence of substantial bias was found in Patient demographics, cardiac arrest related characteristics and major outcomes were predominantly similar between HES matched and unmatched groups, in the linkage apart from location, response time and ROSC at handover."

Although the focus of the study is feasibility, it would be helpful for the authors to discuss what specific OHCA data points can reliably be obtained from registry data and what data points are likely to require active data collection by a study team.

RESPONSE: As suggested by the reviewer, we considered the reliability of different data points and overall strategy as well as feasibility. We added our suggestion in the 4th recommendation as follows: "Therefore, we suggest that trialists use registry data as the main source of all in-hospital data points and active data collection by a study team as an auxiliary approach to collect data for the unmatched patients."

FORMATTING AMENDMENTS (if any)

Required amendments will be listed here; please include these changes in your revised version: